# The Role of Cohesion and Productivity Norms in Performance and Social Effectiveness of Work Groups and Informal Subgroups

**DOI:** 10.3390/bs13050361

**Published:** 2023-04-26

**Authors:** Andrey V. Sidorenkov, Evgueni F. Borokhovski

**Affiliations:** 1Academy of Psychology and Education, Southern Federal University, 105/42 Bolshaya Sadovaya Str., Rostov-on-Don 344006, Russia; av.sidorenkov@yandex.ru; 2Centre for the Study of Learning and Performance (CSLP), Concordia University, 1515 St. Catherine Street West, S-GA-2.126, Montreal, QC H3G 1W1, Canada

**Keywords:** work group, informal subgroup, cohesion, productivity norm, performance effectiveness, social effectiveness

## Abstract

The study addresses the direct and indirect relationship of group cohesion and productivity norm with the perceived performance effectiveness (plan and current tasks implementation and performance success in challenging conditions) and social effectiveness (satisfaction with the group/subgroup and psychological comfort in the group/subgroup) at the levels of work groups and informal subgroups. Thirty-nine work groups from fifteen Russian organizations of different activity profiles, namely services, trade, and manufacturing, took part in the study. The vast majority of them were characterized by relatively low task interdependence. Within the work groups, informal subgroups (from one to three per group) were identified. The cohesion of groups and subgroups was positively and significantly stronger associated with their social effectiveness than with performance effectiveness. The cohesion of subgroups was also indirectly related to social effectiveness of the work groups, i.e., this association was mediated by the subgroup social effectiveness. The index of productivity norm was positively related to perceived performance effectiveness only at the subgroup level, but not at the group level. The productivity norm of the subgroups was also indirectly related to the perceived performance effectiveness of the groups, i.e., this association was mediated by the subgroup performance effectiveness. The indirect relationship between subgroup productivity norm and group performance effectiveness was more complex when cohesion within subgroups was taken into account.

## 1. Introduction

Throughout the history of studying work groups in organizations, researchers have sought to identify, describe, and understand phenomena and processes that make such groups effective. Particularly, since L. Festinger first defined group cohesion [1], this construct has been actively considered as a possible predictor of group efficiency. Between the mid-1960s and the early 1990s, several reviews of research dedicated to the relationships between cohesion and group effectiveness have been published [2,3,4,5,6], drawing rather ambiguous conclusions. Specifically, some questioned the effect of cohesion on group efficiency [4,7], whereas others supported the notion that generally cohesion contributes to group productivity [5]. After the early 1990s, more than 10 relevant meta-analyses have been conducted and published [8,9,10,11]. They provided a more differentiated picture of the cohesiveness–efficiency relationship as a number of specific variables were taken into account, namely type of groups, type of tasks, level of analysis of cohesion, criteria and methods for assessing group effectiveness, etc. In addition to cohesion, researchers paid attention (although much less frequently) to group norms, such as performance norms [12]; norms of conduct [13]; cooperative norms [14]; sociability norms [15]; collectivistic norms [16]; and risk-taking norms [17], all as predictors of performance of work groups. Among them, productivity norms (also known as performance or task norms) should be especially highlighted. This type of norm was considered not so much as an independent predictor of group effectiveness, but in combination with group cohesion [7,18]. There are several circumstances that draw attention. First, the relationship of cohesion and productivity norms is studied primarily with group performance (task-related) effectiveness, whereas the data regarding the relationship of these characteristics with group social effectiveness (e.g., satisfaction of the members with their group and psychological comfort within it) are notably limited.

Second, the relationship of cohesion and productivity norms with performance effectiveness is considered, as a rule, at the level of the entire work group. However, in the vast majority of such groups, informal subgroups emerge [19], which are collective actors, each capable of influencing, to a greater or lesser extent, the life of the entire in-group. Similarly, to the entire group, each subgroup is characterized by a certain degree of cohesion and has its own norms, values, interests, etc. Subgroups also demonstrate various levels of effectiveness. That is, members within the informal subgroup join their efforts and seek to coordinate actions in the course of the work group’s activities. It is easier and more efficient for them to act in the context of the informal subgroup than in the context of the entire group, especially a large one. For example, employees included in a subgroup discuss work problems and methods for completing group tasks, actively help each other, work together to achieve a common goal, etc. (this is what pertains to performance effectiveness). Specifically, within a subgroup (more than within the group as a whole), members may experience a certain degree of psychological comfort and satisfaction with their subgroup (this is what is referred to as social effectiveness). 

Therefore, it is only logical to study the role of different characteristics of subgroups (including cohesion and productivity norms) in determining their respective effectiveness. Moreover, the effectiveness of a group as a whole may depend on the socio-psychological characteristics, activity, and effectiveness of informal subgroups within it. This is an important line of research that has been left rather in the shadows for many years.

The purpose of the current study is two-fold: To undertake a comparative analysis of the relationship of cohesion and productivity norms with two types of effectiveness—performance effectiveness and social effectiveness—at the levels of work groups and informal subgroups (included in these groups);To explore the direct and indirect connections between the cohesion and productivity norms of informal subgroups, on the one hand, and the two kinds of work group effectiveness, on the other hand.

This study contributes to (a) expansion and more refined systematization of ideas about group effectiveness, namely differentiation of efficiency into two types and selection of criteria (indicators) within each ne of them; (b) understanding of informal subgroups as collective actors within the in-group, specificallythose that demonstrate collective effectiveness depending on their certain characteristics (in particular, cohesion and productivity norms); and (c) understanding of the role of informal subgroups (their characteristics and effectiveness) in the effectiveness of the work group as a whole. 

The literature review that follows consists of the two sections. In the first section, we briefly outline the micro-group theory as a conceptual framework for our research. In the second section, we analyze various points of view on such closely related concepts as “group effectiveness”, “group performance”, and “group efficiency” and propose our own systematization of these concepts, specifically highlighting two types of effectiveness, performance effectiveness and social effectiveness, as well as certain criteria to identify each of them. Furthermore, we define informal subgroups and show how they are different from conditional subgroups (the latter are studied much more extensively, whereas the former are rather underrepresented in the research literature) in small groups. Then, we review the results of the meta-analyses that addressed the cohesiveness–efficiency relationships and, based on this, formulate hypotheses about the relationship of cohesion with performance effectiveness and social effectiveness at the levels of both a work group and an informal subgroup. Furthermore, in Section 4, the question of the relationship between the productivity norms and the two types of effectiveness of work groups and informal subgroups is discussed, followed by formulation of the corresponding hypotheses. Finally, we consider the problem of the interaction effects of cohesion and performance norms on performance effectiveness. 

## 2. Theoretical Framework

Conceptually this study is grounded in the micro-group theory [20,21]. This theory developed a set of ideas about (1) three levels of group activity (small group, informal subgroup, and individual) and the connections among them that could be either single-level (e.g., individual—individual, subgroup—subgroup, and group—group) or inter-level (e.g., individual—subgroup, subgroup—group) connections; (2) direct and indirect connections among actors (levels); (3) functions and modes of each level of group activity; (4) types of informal subgroups; (5) contradictions as a universal source of self-change of collective and individual actors; and (6) processes of integration–disintegration as a universal mechanism of self-change of actors, etc. Based on some general ideas of this theory, conceptual models of group phenomena have been further proposed, e.g., intragroup conflict [19], intragroup trust [22], as well as group effectiveness [23]. 

This study used some basic assumptions of the micro-group theory about informal subgroups as collective actors, namely that subgroups (a) have the same characteristics as the group as a whole, and some characteristics (for example, cohesion, trust) within subgroups are stronger, while others (for example, competition, conflict) are weaker than in the group as a whole; (b) perform some functions in relation to the group as a whole, in particular, the function of contributing to completion of the joint in-group tasks (i.e., the subgroup members join forces to accomplish the tasks of the group activity together). In addition, we relied on the idea of the two types of effectiveness—performance effectiveness and social effectiveness—each of which has two aspects: potential and actual. 

## 3. Literature Review and Hypothesis Development 

### 3.1. Types and Criteria of Group Effectiveness

The literature on small groups features several closely related concepts: “group (team) effectiveness”, “group (team) performance”, and “group (team) efficiency”. Moreover, these constructs are not always defined consistently, as researchers sometimes interpret them differently, outline their different aspects, and single out different criteria for identifying them. This creates some terminological confusion. For example, Hackman [24] proposed three criteria for evaluating group effectiveness: the actual result (outcome) of the group activity and its compliance with (or excess of) the specified/expected standards; the group state of readiness and ability of its members to continue working together; and the prevalence of satisfaction (over frustration) with group experiences among its individual members. Gladstein [25] distinguished three dimensions of group effectiveness: performance, satisfying needs of the group members, and the ability of a group to exist for a relatively long period of time. Similarly, Sundstrom et al. [26] believed that group effectiveness has two aspects: performance and viability (i.e., the willingness of members to continue to work together), which ensures the future of the group as a structural unit. However, Cohen and Bailey [27] took a slightly different view of group effectiveness. In addition to performance, they also highlighted attitudes (e.g., job satisfaction, trust) and group behavior (e.g., being late for work, absenteeism). In all these approaches, it is quite clear that the construct “group effectiveness” is wider than the construct “group performance”.

In a broad sense, the term “group (team) performance” is defined as a degree of achieving by a group its goals, accomplishing its mission [28], and is more specifically depicted in terms of performance quality and quantity (productivity) and customer satisfactions with the product/services of the group [29]. Some researchers consider this construct as the level of meeting by a group some pre-established standards for quality, cost, and timeline [30,31] and describe it using variables of effectiveness and efficiency [32]. In this sense, the term “group performance” is broader than the concept of “group effectiveness”. Indeed, the term “performance” is often interpreted as a global (holistic) assessment of the overall group activity that encompasses indicators of effectiveness, success, quality, adequacy, etc., which makes it rather vague and its meaningful interpretation extremely challenging.

The term “group (team) efficiency” [32,33] is less common. It reflects the expediency, the degree of optimal use of material, labor, and other resources in the process of implementing a certain activity, as well as compliance with the work schedule, budget, etc. This characteristic of group activity also reflects that the group work is performed as competently and quickly as possible.

There are several important points to pay attention to. First, some aspects (criteria) of group effectiveness that appear in the research literature are debatable. For example, attitudes are psychological phenomena that manifest themselves primarily at the individual level, whereas group effectiveness is, by definition, a characteristic of a group as a collective actor. Yet, many researchers consider attitudes (for example, trust, group identification, and commitment) as predictors of group performance or variables that mediate its association with some independent variables [34,35,36,37]. Therefore, the question arises if attitudes can be considered as one of the aspects of group effectiveness. Employees’ behavior associated with violation of job discipline can hardly be attributed to group effectiveness. It is behavior of individuals, but the one that, however, can affect (be reflected in) group effectiveness, but cannot be reduced to (equated with) it. Some researchers study extra-role (citizenship) behavior, similarly to the case of attitudes, as an antecedent of group effectiveness [38], whereas others consider it to be a mediator in (i.e., a process that helps shape) the complex relationships between “inputs” and “outputs” of group activity [39]. 

Second, some aspects (criteria) of group effectiveness are substantively different in the context of various group functions. We assume that a work group carries out two major categories of functions: (1) functions that serve (oriented toward) the organization in which the group is included and (2) functions in relation to its members [20,21]. The first category involves the function of establishing and maintaining stability of the organizational environment, since work groups are the basic structural units of the organization, and the task/goal-oriented function, e.g., manufacturing products, completing projects, performing trade, providing services, conducting financial transactions, managing technological or social systems, and maintaining public order. In the context of such (organizational) functions, group effectiveness is described by questions such as What contribution has the group made to the organization? With what costs (time, resources, etc.) did the group do it? In our opinion, this category of functions corresponds to the construct of “performance effectiveness”. This type of effectiveness corresponds to the following criteria: (a) task effectiveness (reflects the degree of compliance of the results achieved by the group with the tasks set, for example, the amount of work performed, the number of completed and not completed tasks, the accuracy of task completion, and the results for the current period relative to the results for previous periods); (b) process effectiveness (reflects the temporal and/or structural components of the work, for example, meeting the deadlines for completing tasks or projects, solving emerging problems or eliminating/mitigating emergency situations, and maintaining the sequence of all operations in accordance with given standards); (c) quality effectiveness (reflects how both the results and work processes meet quality standards, for example, the quality of services provided or products produced); and (d) efficiency or cost-effectiveness (reflects the ratio of result achieved to the associated costs, or of planned and actual costs). Systematizing criteria of performance effectiveness (at least the first three) in such way can also be extended to informal subgroups within a work group. This is justified, since subgroups perform similar functions in relation to their work groups, including the function of contributing to completion of the joint in-group tasks (i.e., the subgroup members join forces to collectively accomplish the tasks of the group activity). Thus, there may be four group (subgroup) performance effectiveness criteria, each of which may contain certain specific objective or subjective indicators.

The other category of functions (oriented toward the group members) includes such functions as informing, training, promoting, and supporting realization of members’ personal goals and social needs, protection from external and internal threats, behavioral regulation, and adaptation [20,40]. Similar functions are also performed by an informal subgroup in relation to its members. Moreover, the subgroup, in comparison with the entire group, is able to perform some of these functions even more successfully, since it is smaller in number of members, and is capable of building stronger trust among them and, subsequently, enjoy higher levels of cohesion, etc. In the context of such functions, group effectiveness is well described by the main question: What did the group do for each of its members? This category of functions corresponds to the construct “social effectiveness”. This type of effectiveness may have such indicators as satisfaction of the group/subgroup members with their respective unit, their psychological comfort within the group/subgroup, group/subgroup viability, etc. Importantly, these indicators can only be subjective.

So, in this study, we will consider two types of effectiveness of work groups and informal subgroups included in these groups: *performance effectiveness* (PE) and *social effectiveness* (SE).

### 3.2. Informal Subgroups in the Work Group

A real (actual) informal subgroup (micro-group) is defined as a subset of group members, which is characterized by a unique form or degree of interdependence [41], or a set of group members who are united on the basis of one or more psychological attributes that are more common and meaningful to them and who have close ties among themselves compared to that of other members [20]. In this sense, a real informal subgroup differs from a conditional subgroup (artificially identified) in a small group identified by the researcher(s) either: (a) on some formal criterion or randomly in a laboratory experiment [42,43,44]; (b) based on so-called “faultlines” associated with variations in group composition defined by simultaneously shared several (at least two) specific demographic or other formal characteristics of group members [45,46,47]. Moreover, the construct “informal subgroup” differs from the commonly used concept “subgroup” which often refers to a real formal group (for example, a team), just included in a larger formal group (for example, a workshop or a department) or belonging to a large social category defined based on a certain socio-demographic quality such as gender, race, etc. [48,49].

Relatively stable informal subgroups are formed in different work groups/teams [19,50,51] and perform certain functions in relation to their members and to the group as a whole [20]. Informal subgroups are collective actors within another collective actor: the work group. General group tasks (especially relatively autonomous ones) are often performed by means of the combined efforts of individual members within informal subgroups (e.g., discussion of emerging problems and ways to solve them, coordination of actions, and mutual assistance). That is, group tasks, especially divisible ones [4], are carried out through the activities of each subgroup, and not necessarily through the totality of the actions of all members of the group. It is worth mentioning that mathematical modeling of team performance has shown that dividing teams into subgroups in conjunction with investing in ICT can significantly increase the effectiveness and performance potential of the organization [52]. There are essential prerequisites for successful completion of group tasks by subgroups. For example, it has been shown that within subgroups, integration among individual members and of members with a subgroup is more pronounced (in the form of cohesion, interpersonal and subgroup identification, interpersonal and subgroup trust, etc.) than in the group as a whole [23]. This significantly contributes to mutual understanding and higher coordination of actions of members within subgroups. Thus, interactions and joint performance of the group tasks are stronger within many informal subgroups than in the context of the entire group. Therefore, the effectiveness of the work group as a whole may depend on the effectiveness of various informal subgroups within it, which, in turn, depends on the productivity norms and cohesion within these subgroups (i.e., depicting a hypothetical pathway: cohesion and productivity norms of the subgroup—effectiveness of the subgroup—effectiveness of the entire group).

### 3.3. Relationship between Cohesion, Productivity Norms and Group/Subgroup Effectiveness

#### 3.3.1. Cohesion and Effectiveness of a Group/Subgroup

In the most general terms, we understand cohesion as the psychological unity of the group, perceived from within (from the members’ point of view) or from the outside (from the side of non-members of the group), according to one or more attributes of relations: (a) the relationships of actors (individuals and subgroups) to the group tasks and the process of joint activities and (b) the relationships between actors (individual—individual, subgroup—subgroup). However, we acknowledge that there are differing views on aspects of group cohesion as a multidimensional construct [10,53]. 

If we turn to the relevant meta-analyses, we see that they analyzed different types of groups: student, industrial, managerial, etc. [9,10,54,55], from different fields of activity: sports [11,56], military [57], labor—design and development, manufacturing, services [8,58,59], entrepreneurial [60], as well as virtual teams [61]. All these meta-analyses considered group PE but did not take into account group SE. Significant positive relationships were found between cohesion and PE, but the magnitude of the corresponding effect sizes varied. Subsequently, the authors drew attention to some important points regarding the strength of these type of relationships. First, the effect size depended on the level of analysis (based on the method of measurements) of cohesion: individual or group [9]. In this regard, the authors of this meta-analysis argued in favor of assessing cohesion primarily at the group level. In the current study, we followed this suggestion and assessed group/subgroup cohesion, respectively, at the group and subgroup levels.

Second, there was also an observation that the cohesion–PE relationship is moderated by some variables, such as degree of task interdependence [9], group size, professional occupation, and group life span [61]. For example, cohesion and PE were more closely related when task interdependence is high than when task interdependence is low [9]. In our study, we predominantly assessed the groups with relatively low task interdependence. Following the logic of the results of these meta-analyses, in such groups, cohesion would not be significantly associated with group PE. However, it may be significantly related to group SE. The same applies to informal subgroups: cohesion of such subgroup should be more strongly related to their SE than to their PE. In addition, cohesion within informal subgroups may be related to the SE of the corresponding in-groups indirectly, i.e., this relationship can be mediated by the respective effectiveness of the subgroups as follows: subgroup cohesion–subgroup SE–work group SE. This assumption is based on the fact that cohesion and SE within informal subgroups are much stronger than cohesion and SE of work groups [23].

Third, the researchers asked if the relationship between cohesion and PE of the groups depends on the cohesion type. Specifically, many authors believe that cohesion is a multidimensional construct, and therefore it has become customary to consider models that distinguish among various aspects of cohesion, for example, interpersonal attraction, commitment to the task, and group pride [10], or individual attractions to the group-task, individual attractions to the group-social, group integration-task, and group integration-social [53,62]. However, over time, there has been no conclusive evidence for such patterns in support for these models. For instance, the subsequent research on the second model, instead of a four-factor structure, either discovered a two-factor one, integration-task and integration-social [63], or failed to confirm even a two-factor solution [64]. Nevertheless, a meta-analysis by Carron et al. [56] showed that social cohesion (integration-social), task cohesion (integration-task), and their combined score were significantly associated with PE, and the differences among these connections were not statistically significant. Considering this, in the current study, we evaluated the integral indicator of cohesion obtained on the basis of the social cohesion and task cohesion indicators.

Fourth, the cohesion–PE relationship may depend on how PE is measured. One meta-analysis found that the same association pattern of group cohesion and PE was observed regardless of whether PE was assessed through self-reports or by objective measures of actual performance/behavior [56]. In another [54], a stronger correlation between cohesion and PE was found when the latter was measured as a behavior associated with the optimal use of resources (efficiency) than when it was measured as an outcome (effectiveness). Our study measured the perceived PE of the group as a whole and the PE of each informal subgroup separately, both (a) as an outcome and (b) as a process (behavior) under difficult circumstances. We also assessed the SE (by two indicators) of the group as a whole and of each subgroup within it.

It should be noted that there are very few studies, e.g., [65], that have studied the relationship of cohesion with SE (for example, with satisfaction of the group members, psychological comfort of the group members, and promotion by the group of professional and/or personal growth of its members). However, SE should not be ignored, as it is important for the psychological well-being and functioning of the group members. We also assume that SE may affect PE. In addition, we could not locate research that would be devoted specifically to the study of the relationship of cohesion within informal subgroups with their PE and SE. So, we formulate the following research hypotheses:

**Hypothesis 1a (H1a):** *Work group cohesion (with relatively low task interdependence) is positively and more strongly associated with SE than with perceived PE*.

**Hypothesis 1b (H1b):** *Cohesion of informal subgroups is positively and more strongly associated with their SE than with perceived PE*.

**Hypothesis 1c (H1c):** *The association of cohesion of informal subgroups with SE of work in-groups is mediated by SE of subgroups (Figure 1)*.

#### 3.3.2. Productivity Norms and Group/Subgroup Effectiveness

Productivity norms (in some sources also referred to as norms for productivity, performance norms, and task norms) and effectiveness of the group/subgroup are understood as collective expectations of the group members regarding the performance of joint activities and their respective results [66], or as collective opinions about the degree of efforts made by the group members to perform their principle activities [67]. In our opinion, the productivity norms of a group/subgroup are standards for performing activities accepted in the group and shared by the majority of its members and imply a certain degree of intensity and amount of the work performed. These norms are very frequently considered using sports teams as an example. Studies of such teams have shown a positive relationship of productivity norms with team success [68] and satisfaction with team performance [69]. However, the strength of the relationship between productivity norms and PE at the work group level may depend on the measure of task interdependence. Specifically, in work groups with relatively low task interdependence, the relationship among the variables we are interested in supposedly will be weaker than in groups with high task interdependence. Moreover, the positive relationship between productivity norms and PE at the informal subgroup level may be much stronger than the relationship between these variables at the work group level. It is quite likely that the productivity norms in the informal subgroup, compared with the work group as a whole, are accepted by its members to a greater degree, and monitoring and observing them is easier performed at the subgroup level. This is rooted in the fact that the informal subgroup, compared to the work group, is smaller in number, more coherent (has stronger internal integration), and is more capable of controlling behaviors of its members. Based on this and taking into account that informal subgroups are indeed collective actors with their unique contributions to effectiveness of the entire group, it is plausible that productivity norms of the informal subgroups can be connected—directly and indirectly—with PE of the corresponding work in-groups. Moreover, the indirect connection/relationship can be mediated by the respective subgroup performance as follows: subgroup productivity norm–subgroup PE–group PE. Therefore, we formulated the next set of research hypotheses.

**Hypothesis 2a (H2a):** *The relationship between productivity norm and perceived PE is positive and stronger at the informal subgroup level than at the work group level with relatively low task interdependence*.

**Hypothesis 2b (H2b):** *The productivity norm of informal subgroups is positively associated with perceived PE of work groups*.

**Hypothesis 2c (H2c):** *The association of informal subgroups productivity norm with perceived PE of work groups is mediated by PE of respective subgroups (Figure 1)*.

From now on, we use the term “norm” in its singular term as in the current study only one indicator is employed and analyzed.

#### 3.3.3. Interactive Effect of Cohesion and Productivity Norms of Informal Subgroups on the Effectiveness of Work Groups

The early laboratory experiments showed that group productivity depends on group cohesion and, at the same time, group induction, i.e., on the influence the group imposes on its members in order to increase/decrease productivity [70] and group standard, i.e., “the expressed and shared attitudes of the group members toward their task” [71]. In one experiment, productivity was found to increase significantly in both high- and low-cohesion groups if they were also high in induction, whereas under low-induction conditions, high-cohesion groups were less productive than low-cohesion groups [70]. In another (replication) experiment, it was found that high- and low-cohesion groups did not differ in productivity before the group standard was introduced to the experiment [71]. High cohesion and high standard contributed to increased productivity while low standard and low cohesion were associated with low productivity. These results clearly suggest that the change in productivity is a function of the change in group induction/standard, and not in group cohesion. Later, Stogdill [7] analyzed thirty-four studies conducted with different groups and found that cohesion was positively associated with performance in twelve groups, negatively associated with performance in eleven, and unrelated to performance in eleven others. In his opinion, the important factor that determines/influences the relationship between cohesion and performance is “drive”. Drive is defined as the “…degree of group arousal, motivation, freedom, enthusiasm, or esprit… the intensity with which members invest expectation and energy on behalf of the group.” [7] (p. 27). In a sense, drive is associated with task motivation, competition, frustration, stress, etc. If group cohesion is high and drive is high, performance is increased. When cohesion is high and drive is low, performance is low or negatively impacted. When cohesion is low, high drive groups will outperform low drive groups. This point of view has continuously been strengthened in the literature, and over time, the term drive was replaced by the concept of “productivity norm”. However, it is obvious that the constructs “drive” and “group standard”/“productivity norm” differ significantly from each other, as evident from their respective definitions given above.

A study of sports teams found that teams with higher norms (including task-related norms) and higher social cohesion had a higher perceived team PE (as reflected in the degree of efforts made by the team members), whereas the lowest efficiency was observed in teams with lower social cohesion and higher norms [72]. Interestingly, task-related norms and task cohesion were not associated with team PE as perceived by its members. In teams from family service agencies and in military units, the combination of high cohesion and high task performance norms, compared to any other combination, leads team leaders to perceive higher group PE such as quality and accuracy of work and optimal unit operations [18]. However, within these groups, there was variation in the effects of other combinations of cohesion and productivity norms. Apparently, the overall picture of the interaction effects (high vs. low cohesion in combination with high vs. low productivity norms) on group PE is not so unambiguous. It could be suggested that the effect of interaction between cohesion and productivity norms depends on the content of joint activity and the degree of interdependence of group tasks.

The study of interactive effects of cohesion and productivity norms at the work group level is not the focus of our study. However, applying the traditional for this matter logic to informal subgroups, we focus attention on the role of the combination of their cohesion and productivity norms in PE of the corresponding work in-groups. Based on H2c above, we formulated a hypothesis that reflects the model of this type of moderated/mediation:

**Hypothesis 3a (H3a):** *The association of productivity norm and perceived PE of informal subgroups is moderated by cohesion of these subgroups (Figure 2)*.

**Hypothesis 3b (H3b):** *The indirect effect of informal subgroups productivity norm on work groups PE through subgroups PE is enhanced by subgroups cohesion (which moderates the productivity norm–PE of the subgroup relationship), so that this indirect positive effect is expected to be stronger in high-cohesion subgroups (Figure 2)*.

## 4. Method

### 4.1. Sample

The study was conducted with 39 work groups from 15 Russian commercial enterprises and state institutions of different activity profiles, namely services (both social and commercial), trade, and manufacturing. The study was conducted in those organizations whose management agreed to provide access to (the leadership of some of the organizations we approached refused to participate). We deliberately took the path of assembling a wide range of organizations in order to make the sample more diverse and representative. In each of these organizations, we studied 1–6 groups whose employees agreed to participate in the study. 

The total number of participants was 349. Group size varied from 5 to 17 members (M = 8.94, SD = 2.91). The sample included 51.6% women and 48.4% men aged from 19 to 62 (M = 37.9). Fifteen groups were homogenous in terms of gender, of which ten consisted entirely of men and five of women. The vast majority of groups could be characterized as having relatively low task interdependence.

In all examined groups, the study identified sixty-nine informal subgroups, from one to three per participating formal work group. More specifically, in 66.7% of the groups, two or more informal subgroups were identified. These informal subgroups were predominantly dyads (50.7% of all subgroups), informal triads were represented by 26.1% of the sample, followed by noticeably fewer number of subgroups composed of four or five members (11.6% of each type).

### 4.2. Measures

The study was conducted using the GROUP PROFILE software package [73], which included individual questionnaires designed to assess cohesion, productivity norm, and group effectiveness, as well as an integrated with them algorithm (formalized data processing algorithm) for identifying informal subgroups within formal work groups.

The cohesion of the work group and the informal subgroups was measured using the task cohesion (TC) and social cohesion (SC) questionnaire. It consists of the two respective subscales with five items (with the negatively framed statements) in each of them, for example, “Co-workers do not tend to work together” (TC), “Group members do not seek to consider the interests of each other” (SC), etc. Each statement is to be evaluated using a 7-point scale, from “absolutely true” to “absolutely false”. The questionnaire consists of two parts: (1) “The group as a whole” (to assess the cohesion of the entire work group) and (2) “Among those with whom I maintain close relations” (to assess the cohesion of the subgroups). Cronbach’s α indices were 0.763 and 0.753 for the TC and the SC subscales, respectively (at the group level). Taking into account rather high correlation between subscales (0.84) and that, according to the meta-analytical findings, there is lack of significant differences between the associations of similar types of cohesion with performance [56], it was decided to use an integral indicator for the two subscales.

The productivity norm of work groups and informal subgroups was assessed by means of a questionnaire, which contains five items formulated as proverbial sayings that reflect such job-related standards as the pace and amount of the work performed (for example, “Go slower, reach further”, “One can’t accomplish everything”). This questionnaire, like the previous one, consists of two parts, for assessing productivity norm of the work group and each subgroup within it, respectively. The same 7-point evaluation scale applies, and the Cronbach’s α value was 0.834.

Due to problems with consistent availability of objective data on PE across participating groups, we had to limit ourselves to its subjective indicators. To assess PE of the work groups and informal subgroups, a questionnaire of the same name was used. It includes two subscales: “Plan and current tasks implementation” (P&T) and “Performance success in challenging conditions” (SCC) with three items (also framed in negative/reversed terms) in each, e.g., “The group often does not implement the pre-set plan according to major performance indicators” (P&T subscale) or “The group does not know how to timely solve new tasks or complex problems” (SCC subscale). The first subscale assesses the overall performance of the group/subgroup, and the second the process of performing work in special conditions. Each item was evaluated on a seven-point scale (as described above), separately by the group members (P&T-M and SCC-M) and by the group leader (P&T-L and SCC-L). Group members only evaluated the effectiveness of the group as a whole. After employees completed the questionnaire, the group leader assessed effectiveness of the entire group and each subgroup in a special module, which offered the same items alongside the information about the number and composition of the identified informal subgroups. Cronbach’s α values were 0.851 and 0.927 for the P&T and SCC subscale, respectively.

SE of participating work groups and informal subgroups was measured by another questionnaire on its two subscales: “Satisfaction of the members with their respective group/subgroup” (SG) and “Psychological comfort of the members in their respective group/subgroup” (PC). Each subscale included two negatively framed items, e.g., “I am not satisfied with how the things are going in the group” (SG subscale), or “I feel uncomfortable and/or tense while in the group” (PC subscale). There are also two parts of the questionnaire for assessing SE at the group (i.e., “In the group as a whole”) and at the informal subgroup (i.e., “Among those with whom I maintain close relationships”) levels. Participants evaluated each item on the same seven-point scale. Cronbach’s α values were 0.856 and 0.827 for the SG and PC subscales, respectively.

### 4.3. Procedure

Before participating in the study, all respondents were informed about its objectives. The study was conducted at the respective workplaces, approved by the administration of the relevant organizations and with the consent of individual participants, expressed orally. Participants took turns working on laptops with the GROUP PROFILE computer program. After the data collection was completed, this software automatically processed the results for each group separately. 

### 4.4. Data Analysis

The data analyses included the calculation of the descriptive statistics and Spearman correlation coefficients, the procedure of normalized (conversion to T-scores), regression analysis, as well as the bias-corrected bootstrapping technique. The analytical procedures were carried out using statistical software package *SPSS 23,* including *PROCESS-macro* (Models 1, 4 and 7) within it.

## 5. Results

### 5.1. Cohesion–Effectiveness Relationship

Table 1 shows the descriptive statistics and correlations among the variables included in testing H1 and H2. Since rank correlation is a non-parametric analytical method, no normality test is required for the variables to be examined. Work group PE indicators were measured separately based on the scores of the group members and the group leaders, while informal subgroup PE indicators were measured on the leaders’ scores only. According to H1a and H1b, cohesion is more strongly associated with SE than with PE at the work group and informal subgroup levels, respectively. It was found that the cohesion of both work groups and subgroups does not significantly correlate with either of the two indicators of PE, but is significantly positively related to both indicators of SE. We compared the correlation coefficients presented in Table 1 using the Fisher z-transformation. That is, we compared the cohesion–PE correlations (for each indicator of this efficiency type) with the correlation of cohesion–SE (for each indicator of this efficiency type). At the group level, all differences in correlations are statistically significant (in seven compared correlations, *p* < 0.001, and in one, *p* < 0.01). At the subgroup level, three of the four comparisons found significant differences in correlations (*p* < 0.05, *p* < 0.01, and *p* < 0.001), but not significant differences in one (*p* > 0.05): between the cohesion–PE association (by the indicator “plan and current tasks implementation by the subgroup”) and the cohesion–SE association (by the indicator “satisfaction with the subgroup”). Therefore, H1a and H1b are supported.

H1c suggested an indirect relationship between the cohesion of informal subgroups and the SE of work in-groups, mediated by the corresponding effectiveness of subgroups. The bias-corrected bootstrapping approach (with 5000 resamples) was used to analyze the effects of mediation. This approach, unlike many other widely used methods, does not require for a sample to be normally distributed [74]. The procedure was performed using PROCESS macro [75], Model 4 of the SPSS 23 software package. Variables were mean centered prior to the analysis. If the zero does not fall within the confidence interval, the mediation effect is significant. The results of these analyzes are presented in Table 2. As shown there, SE (on the “psychological comfort” indicator) of informal subgroups mediates the association (*b* = 0.061, 95% CI [0.013, 0.121]) of subgroup cohesion with SE (on the corresponding indicator) in work groups (*b* = 0.061, 95% CI [0.013, 0.121]). No significant indirect effect of the “satisfaction with the subgroup/group” indicator was found. Subsequently, H1c has not been fully supported, as we expected to detect the mediation effect of cohesion on both SE indicators. However, this hypothesis is still supported for one SE indicator.

### 5.2. Productivity Norm–Effectiveness Relationship

H2a suggests that a positive relationship between productivity norm and perceived PE is stronger at the informal subgroup level than at the formal work group level. Table 1 shows that the productivity norm of informal subgroups is positively and significantly correlated with only one of the two indicators of the corresponding PE (i.e., “plan and current tasks implementation”, as assessed by the group leader). No significant correlation of this norm with the indicator “performance success (of the subgroup) in challenging conditions” (according to the leader) was found. In turn, the productivity norm of the work groups does not have a significant relationship with any of their PE indicators (neither according to the members’ nor to the leader’s assessment). Additionally, comparison of correlations between the productivity norm and the indicator “plan and current tasks implementation” (according to leader’s assessment) both at the level of groups and subgroups showed that they differed significantly (*Z* = −2.198, *p* = 0.014). So, H2a is supported, but not in full. That is, the productivity norm of informal subgroups, in comparison with the same norm of groups, has a positive significant and stronger association with one of the two PE indicators (according to the leader’s assessment).

We also hypothesized that the productivity norm of informal subgroups is positively associated with the perceived PE of the work groups that include these subgroups (H2b). We tested this hypothesis through regression analysis (Table 3), with all variables normalized (conversion to t-scores). A significant positive relationship was found between subgroup productivity norm and (a) both group PE indicators (assessed by group members) and (b) one indicator of group PE (“performance success in challenging conditions”), as assessed by the leader. Therefore, H2b is largely supported.

H2c states that the productivity norm of informal subgroups is indirectly related to perceived PE of work in-groups, namely that this connection is mediated by perceived PE of the informal subgroups. The bias-corrected bootstrapping approach was used to analyze the effects of the mediation (Model 4). Table 4 shows that the relationship between the productivity norm of subgroups and the PE of work groups (as perceived by their leaders) is significantly mediated by PE of subgroups for the “plan and current tasks implementation” indicator (*b* = 0.096, 95% CI [0.017, 0.187], zero does not fall within the confidence interval). For another indicator “performance success in challenging conditions”, there is no significant mediation effect. So, H2c is supported for one of the two indicators only. 

### 5.3. Interactive Effect of Cohesion and Productivity Norm on PE of Group/Subgroup 

In H3a, we hypothesized that the cohesion of informal subgroups moderates the relationship between their productivity norm and perceived PE. The moderation effect was analyzed using a bias-corrected bootstrapping approach (Model 1). It was found that the cohesion of subgroups significantly moderates the relationship of their productivity norm with the “plan and current tasks implementation” indicator (*b* = 0.036, *p* < 0.05, 95% CI [0.016, 0.090]). Moreover, with high cohesion of subgroups, in contrast to low cohesion, the positive effect of the productivity norm in relation to this PE indicator is enhanced (Table 4). Cohesion does not significantly moderate the association of productivity norm with another subgroup PE indicator, “performance success in challenging conditions” (*b* = 0.022, *p* > 0.05, 95% *CI* [−0.001, 0.085]). In other words, hypothesis H3a is supported with respect to one of the two informal subgroup PE indicators.

We also hypothesized an indirect effect of the productivity norm of informal subgroups on the PE of work in-groups through the corresponding subgroup effectiveness, which is enhanced by subgroup cohesion (H3b). We tested H3b as the full model with Hayes’s Model 7 PROCESS macro, which predicted a moderated mediation model with moderation in the first stage, where the independent variable affects the mediating variable. We used the mean plus one standard deviation to represent a strong subgroup cohesion and the mean minus one standard deviation to represent a weak subgroup cohesion. The results of this analysis are presented in Table 4. We found support for this indirect effect for the PE indicator of “plan and current tasks implementation”, as follows: (a) when subgroup cohesion was strong, the mediating role of subgroup PE for the association between subgroup productivity norm and work group PE was significant (95% CI [0.043, 0.306]); and (b) when subgroup cohesion was weak, the mediating role of subgroup PE for the same association was not significant (95% CI [−0.309, 0.051]). Hayes [76] proposed an index of moderated mediation as a formal inferential test of whether the moderated mediation model is statistically different from zero. The results showed that the index excluded zero in the 95% confidence interval (*B* = 0.013, *SE* = 0.008, 95% CI [0.004, 0.036]), i.e., was statistically significant. However, no indirect effect was found for another PE indicator, “performance success in challenging conditions” (*B* = 0.010, *SE* = 0.010, 95% CI [−0.001, 0.038]).

## 6. Discussion

As it has been repeatedly shown, the internal cohesion of small groups and informal subgroups is positively associated primarily with their SE than with PE. This is convincingly demonstrated in work groups with relatively low task interdependence. Such groups do not require a high degree of coordination (that largely depends on group cohesion) to effectively carry out the work at hand. If our sample consisted predominantly of groups with high degree of tasks and/or outcomes interdependence, then with a high probability we would have found a significant positive relationship between work group cohesion and its PE. This assumption is based on some meta-analytical results [9,58]. 

However, group cohesion, as the psychological unity of its members, is an important prerequisite for satisfaction of the members of the group/subgroup and their psychological comfort within the group/subgroup. At the same time, we recognize a possible reciprocal connection, i.e., the inverse effect of SE on group/subgroup cohesion. That is, the more members are satisfied with their group/subgroup and feel more comfortable within it, the more grounds there will be for their unity. It is worth noting that the connection from performance to cohesion may produce more direct effects than that from cohesion to performance [10].

It is also important to pay attention to the fact that the cohesion of informal subgroups is indirectly related to such an SE indicator of work groups as psychological comfort of their members. This relationship is mediated by the respective effectiveness of the subgroups. That is, the stronger subgroups’ cohesion, the more salient their SE indicator of ‘psychological comfort’ and, as a result, the stronger the corresponding SE indicator of work groups as structural units. Indeed, trust, identification, and cohesion are stronger within subgroups than in the group as a whole [23]. Respectively, we found that the members included in subgroups feel more comfortable within their subgroup than all members in the in-group as a whole (*Z* = −5.124, *p* = 0.000). The extent to which members feel psychological comfort in their subgroups determines how comfortable they will be in the entire work group. Of course, the psychological comfort of members in a work group may also depend on the relationship among informal subgroups. It can be assumed that the more subgroups cooperate and maintain positive relations with each other, the higher will be the psychological comfort of members in the entire work group. On the contrary, if the relations among subgroups are predominantly competing and negative, it may lead to a decrease in psychological comfort, in satisfaction with the group, and even in PE.

We did not find a significant association between productivity norm and PE at the work group level, which is inconsistent with the notion that group productivity norm [68] and group standard [71] are positively related to performance. Probably, the strength of the effect of productivity norm on the group PE may depend on the degree of sharing of that norm by the group members, type of group and the content of the joint activity of its members, as well as on the degree of interdependence of the group task, etc. For example, in work groups with relatively low task interdependence, the performance norm–PE relationship will be weaker than in groups with high task interdependence. In groups with low tasks interdependence, the productivity norm of informal subgroups (which has been shown in this study) and the individual motivation of members not included in subgroups will play a more significant role for the group PE.

In turn, productivity norm of the informal subgroups had a positive connection with the “plan and current tasks implementation” (as assessed by the group leader) indicator of their PE. Moreover, connection is moderated by cohesion within subgroups. The higher the cohesion of subgroups, the stronger the positive effect of productivity norm of subgroups on their plan and current tasks implementation. At the same time, no significant relationship between productivity norm and PE was found at the level of work groups. This fact is probably due to the following circumstances. The productivity norm of subgroups is slightly higher than such a norm of work groups (*Z* = −1.884, *p* = 0.06). Although the degree of differences is not statistically significant, we can still talk about a trend. Another circumstance, as we assume, is that within informal subgroups (compared to the group as a whole), mutual responsibility and mutual understanding among members are stronger. This helps to reduce losses in coordination and motivation within subgroups, which, in turn, favors a positive relationship between productivity norm of subgroups and their perceived PE. Thus, the quite common for the research literature notion about the interactive effect of cohesion and productivity norms on PE at the group level [7,18,72] has not been supported by our study. However, the overall picture of the effects of the interaction between cohesion and productivity norms on group PE is not entirely clear [18]. In our study, this interactive effect was evidently shown only at the level of informal subgroups. 

It was also found that informal subgroup productivity norm had significant direct and indirect relationships with PE of a work group. In the first case, subgroup productivity norm was positively related to both PE indicators perceived by group members and one PE indicator (performance success in challenging conditions), as perceived by group leaders. In the second case, productivity norm of informal subgroups was associated with one PE indicator (plan and current tasks implementation) perceived by the leaders of the work groups, and this relationship was mediated by the corresponding PE indicator of the subgroups. These discovered connections can be explained by the fact that informal subgroups are collective actors, each of which can, to a certain extent, contribute to the results of the corresponding work groups. This contribution depends on productivity norm (mediation) and cohesion (moderation) of the subgroups. In general, the important role of informal subgroups in PE and SE of work groups should be noted. Moreover, we are talking here about the role of certain characteristics (cohesion and productivity norms, specifically) and about the effectiveness of informal subgroups, not only about the degree of perception by members of subgroups, as it has been typically pointed out in the literature [51,77,78]. 

## 7. Conclusions

Researchers are interested in cohesion and productivity norms as predictors of PE of work groups. At the same time, in the research literature, there is rather a lack or even complete absence of studies that address (1) the relationship of these characteristics with social effectiveness of work groups; (2) the role of these characteristics in the PE and SE of informal subgroups that emerge within small groups; (3) the role of these characteristics of informal subgroups in PE and SE of work groups; and (4) the direct and interactive effects of cohesion and productivity norms on the two types of effectiveness simultaneously at the levels of the work group and the informal subgroup within it. This study attempted to fill in the gaps in this problem area and depict directions for the future research.

### 7.1. Theoretical and Practical Implications

This study systematizes and expands scientific understanding of the phenomenon of group effectiveness. The results of this study for the first time demonstrated the relationship of cohesion and productivity norm with two types of effectiveness (SE and PE) at the level of informal subgroups, as well as the relationship of work group cohesion with the two indicators of SE. The results also expand understanding of the role of cohesion and productivity norm in two types of effectiveness of work groups with relatively low task interdependence. Namely, not only group cohesion but also subgroup cohesion is significantly positively related (both directly and indirectly) to SE of work groups. Not so much the group productivity norm as the subgroup productivity norm is related (both directly and indirectly) to PE of the work groups. We have also shown that the indirect relationship between subgroup productivity norm and PE of work group can be more complex when cohesion within subgroups is taken into account. In general, this study relies upon, further develops, and tests the ideas of the micro-group theory about informal subgroups within a work group as collective actors that demonstrate collective effectiveness depending on their certain characteristics, and also contribute to the effectiveness of the group as a whole. 

The study has brought up some practical implications. Knowledge of the association of work group cohesion with its SE, as well as of the association of cohesion and performance norms of informal subgroups with SE and PE of work groups, obtained in the course of this study can contribute to a better understanding of predictors of the two types of work group effectiveness. This will help managers to better realize the role of informal subgroups and their characteristics in the effectiveness of work groups, and influence subgroups accordingly in order to increase the effectiveness of the work group they belong to. However, managers must take into account the degree of interdependence of tasks and the specific working conditions of the group in order to find the optimal solution regarding the role of subgroups in the effectiveness of the entire group. For example, if there is a relatively low interdependence of group tasks, then it is desirable to organize work so that tasks are carried out through the actions of subgroups as collective actors. Here, it is important that the leader contributes to an increase in productivity norms in each subgroup. If the work group, on the contrary, has a high interdependence of tasks, then it is necessary to pay attention to the leading subgroup, if any. In our view, the concept of “leadership” applies not only to an individual, but also to an informal subgroup. In that sense, the “leading subgroup” is the one that enjoys the high status and demonstrates high activity within the work group and imposes stronger influence on the entire work group. Such a subgroup can be the core around which the activities of the other members of the group are organized. It is important that this subgroup is characterized by a high productivity norm and high cohesion.

### 7.2. Limitations and Future Research

This study has some limitations. We studied work groups in which relatively low task interdependence prevailed, and the sample did not include groups with high task interdependence. However, this task characteristic may be a significant moderator of the relationship among the variables in this study. Moreover, the study did not take into account the nature of the interaction (competition, conflict, or cooperation) among subgroups within groups, which could affect the characteristics of the relationship of cohesion, productivity norms, and effectiveness at the group and subgroup levels.

The prospect of further research lies in the development of a broader perspective: the role of informal subgroups in the effectiveness of small groups. This involves the study of (a) the relationship of different characteristics and behaviors of subgroups as collective actors with two types of effectiveness of work groups, which can be moderated by certain tasks and conditions of group work; (b) the effects of attitudes and interactions among subgroups, and between individual subgroups and the group as a whole with respect to group effectiveness; (c) the role in group outcomes of the subgroups status and of influence processes of subgroups; (d) the inclusion/non-inclusion of the manager in an informal subgroup and the type of the leader’s relationship to subgroups in a work group as a condition for success of managerial activities and of the group effectiveness, etc.

## Figures and Tables

**Figure 1 behavsci-13-00361-f001:**
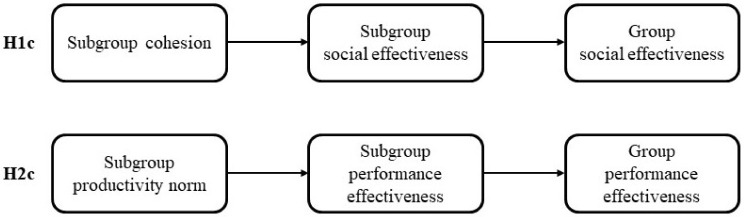
Schematic representations of the associations in H1c and H2c.

**Figure 2 behavsci-13-00361-f002:**
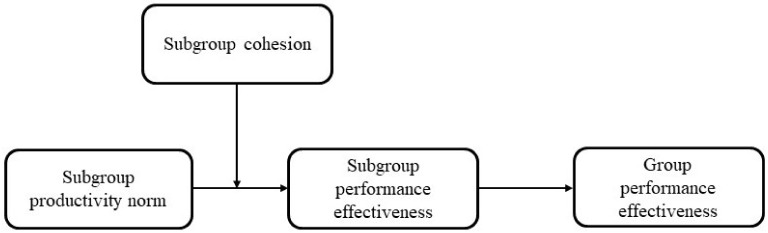
Schematic representations of the associations in H3.

**Table 1 behavsci-13-00361-t001:** Descriptive statistics and Spearman’s correlations among variables at the small group and informal subgroup levels.

Variable:	*M*	*SD*	2	3	4	5	6	7	8
1. Cohesion	41.47 54.06	10.03 9.16	0.67 *** 0.01	0.09 -	0.21 -	−0.06 0.21	−0.08 0.12	0.67 *** 0.30 *	0.84 *** 0.61 ***
2. Productivity norm	19.72 21.62	5.14 5.59		0.04 -	0.24 -	−0.17 0.36 *	−0.03 0.22	0.53 *** 0.30 *	0.64 *** 0.07
3. Plan and current tasks implementation (M)	15.94 -	2.47 -			0.77 *** -	0.36 -	0.26 -	0.01 -	0.14 -
4. Performance success in challenging conditions (M)	15.77 -	2.38 -				0.34 -	0.33 -	0.11 -	0.19 -
5. Plan and current tasks implementation (L)	15.92 16.22	3.95 3.78					0.64 *** 0.58 ***	−0.03 0.47 **	−0.047 −0.25
6. Performance success in challenging conditions (L)	15.73 16.33	3.87 3.74						−0.03 0.69 ***	−0.16 −0.22
7. Satisfaction	7.65 10.21	1.97 1.99							0.71 *** 0.07
8. Psychological comfort	8.71 10.98	2.16 2.06							

*Note*. M—assessment of effectiveness is performed by the group members; L—assessment of effectiveness is performed by the group leader (manager). The numbers above the separating line reflect the results at the work group level, and those below the bar at the level of informal subgroups. * *p* < 0.05; ** *p* < 0.01; *** *p* < 0.001.

**Table 2 behavsci-13-00361-t002:** Indirect Effect of Informal Subgroups’ Cohesion on SE of Work Groups.

Mediation Model (H1c)	*b*	Boot SE	95% Boot LLCI	95% Boot ULCI
*Mediator:* SE of informal subgroups				
Satisfaction	−0.004	0.010	−0.029	0.015
Psychological comfort	0.061	0.027	0.013	0.121

**Table 3 behavsci-13-00361-t003:** Regression analysis: connection of productivity norm of informal subgroups with the indicators of PE of work groups.

PE Indicators	Model Characteristics (H2b)				
	St. β	$\mathrm{Adj}.R^{2}$	BootSE	BootLLCI	BootULCI
Plan and current tasks implementation (M)	0.309 **	0.096	0.116	0.110	0.570
Performance success in challenging conditions (M)	0.402 ***	0.149	0.112	0.217	0.643
Plan and current tasks implementation (L)	0.189	0.036	0.139	−0.097	0.450
Performance success in challenging conditions (L)	0.301 *	0.091	0.138	0.031	0.571

* *p* < 0.05; ** *p* < 0.01; *** *p* < 0.001.

**Table 4 behavsci-13-00361-t004:** Indirect effect of informal subgroup productivity norm on PE of subgroups and work groups.

Models	Estimate				95% CI	
	*b*	*SE*	t	*p*	Boot LLCI	Boot ULCI
Mediation model (H2c):						
Direct effect	0.058 0.134	0.044 0.058	1.295 2.290	0.205 0.029	−0.033 0.014	0.150 0.254
Indirect effect	0.096 0.086	0.043 0.050			0.017 −0.010	0.187 0.185
Moderation model (H3a):						
Low cohesion of subgroups	−0.129 -	0.190 -	−0.681 -	0.501 -	−0.520 -	0.260 -
High cohesion of subgroups	0.451 -	0.127 -	3.550 -	0.001 -	0.190 -	0.711 -
Moderated mediation model (H3b):						
Indirect effect moderated by subgroups’ low cohesion	−0.049 −0.019	0.097 0.125			−0.309 −0.388	0.051 0.109
Indirect effect moderated by subgroups’ high cohesion	0.171 0.140	0.065 0.071			0.043 −0.018	0.306 0.279

PE indicators were only assessed by the team leader. The numbers above the bar represent the “plan and current tasks implementation” PE indicator and those below the separating line the “performance success in challenging conditions” PE indicator.

## Data Availability

The datasets generated during and/or analyzed during the current study are available in the Figshare repository (https://doi.org/10.6084/m9.figshare.22093133, accessed on 14 February 2023).

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
