# Peer review of "The Role of Cohesion and Productivity Norms in Performance and Social Effectiveness of Work Groups and Informal Subgroups"

_behavsci, 2023, doi:10.3390/bs13050361_

Round 1

Reviewer 1 Report

This paper explores an interesting topic such as the role of group cohesion and productivity norms in relation to perceived performance and social effectiveness at the levels of work groups and informal subgroups. In this regard, the consideration of informal subgroups, which are generally underrepresented in the research literature, represents the paper's greatest strength.

The authors demonstrate a good knowledge on the issues at stake, the literature quoted is appropriate. The publication is well structured and adequately reflects the latest work and knowledge on the selected topic. The research results are stimulating despite the limitations pointed out by the authors.  I consider the article to be acceptable in its current form.

Author Response

The authors express their deep gratitude to the referee for the interest to our study and for the thorough analysis of the manuscript.

Thank you!

Reviewer 2 Report

The Work is well documented, however it seems important to me in the introduction to highlight/explain the contribution of the study and to which theory it intends to contribute or expand.

From line 148 “Second, some aspects (criteria) of group effectiveness are substantively different in the context of various group functions.”  until line 177  has a large amount of text where references of this information are not indicated.

In the sample, the sampling technique for the recruitment of study participants is not indicated.

In the method section could indicate the procedures adopted in the analysis of the data.

 Table 1. There are some numbers with comma (e.g., 16,22; 16, 33…) you should correct them (e.g., 16.22; 16.33…)

Line 515 You write “the moderation effect is significant” but it seems that it is “the mediation effect is significant”

 Line 537  You write “H2a was confirmed, but not in full” but it seems that “H2a was supported, but not in full

 Line 559 You write “So, H2c is confirmed for one of the two indicators only.” but it seems that “So, H2c is supported for one..”

 Line 576 You write “hypothesis H3a is confirmed” but it seems that “hypothesis H3a is supported”

Author Response

The authors express their deep gratitude to the referee for the thorough analysis of the manuscript, and hope that the answers to the comments (please, see below), as well as the corresponding adjustments to the manuscript itself are satisfactory.

General comments:

  1. The Work is well documented, however it seems important to me in the introduction to highlight/explain the contribution of the study and to which theory it intends to contribute or expand.

Response: Thank you for your thorough review and valuable comments/suggestions. In the Introduction section (next to the research goals) we briefly explain our understanding of how the study may contribute to the related theory. Also, we added a new subsection “Theoretical Framework” (after the Introduction), which intriduces the micro-group theory as our conceptual framework.

  1. From line 148 “Second, some aspects (criteria) of group effectiveness are substantively different in the context of various group functions.” until line 177 has a large amount of text where references of this information are not indicated.

Response: At the beginning of the indicated paragraph, we referred to the publication, which addresses two categories of group functions. Based on this work we briefly outlined the essence of the first category. Then we identified four criteria of performance effectiveness. All these could be traced back to the literature review presented just above (with the corresponding references). As such, we did not see the need to repeat these references in the paragraph that followed.

  1. In the sample, the sampling technique for the recruitment of study participants is not indicated.

Response: Thank you for the important comment. We added the description of the recruitment procedure.

  1. In the method section could indicate the procedures adopted in the analysis of the data.

Response: We added a subsection “Data Analyses” at the end of the “Method” section.

  1. Table 1. There are some numbers with comma (e.g., 16,22; 16, 33…) you should correct them (e.g., 16.22; 16.33…).

Response: Thank you for pointing our attention to that. It has been corrected.

  1. Line 515 You write “the moderation effect is significant” but it seems that it is “the mediation effect is significant”.

Response: It has been corrected. Thank you.

  1. Line 537  You write “H2a was confirmed, but not in full” but it seems that “H2a was supported, but not in full.

Response: Done.

  1. Line 559 You write “So, H2c is confirmed for one of the two indicators only.” but it seems that “So, H2c is supported for one..”

Response: Done.

  1. Line 576 You write “hypothesis H3a is confirmed” but it seems that “hypothesis H3a is supported”

Response: Done.

Thank you!

Reviewer 3 Report

Good work.  Well-written, well-cited.  Very complex study that was made quite clear.  What is the practical value of this work as the number of variables studied increases?    .  

Author Response

The authors express their deep gratitude to the referee for the interest to our study and for the thorough analysis of the manuscript.

Question: What is the practical value of this work as the number of variables studied increases?   

Response: The applied value of the study lies in the fact that the managers need to pay attention to informal subgroups, as the effectiveness of the subgroups themselves and the work group as a whole may depend on characteristics and activity of subgroups. Moreover, it is necessary to clearly distinguish between two types of efficiency, both of groups and of subgroups that can mutually reinforce or weaken each other.

For example, if a group has relatively low task interdependence, then it is desirable to organize the work of the group so that group tasks are performed through the activity of separate subgroups (with high productivity norms). That is, members within informal subgroups join forces and seek to coordinate actions in the process of performing group tasks. It will be easier and more successful for them to act in this way than in the context of the entire group, especially a large one. If the productivity norms of a subgroup are low, then the manager needs to understand the reasons and strengthen them.

If a work group, on contrary, has a high task interdependence, then it is necessary to pay attention to the leading subgroup, if there is one. Such a subgroup can be the core around which the activities of the other members of the entire group are organized. It is important that this subgroup is characterized by a high productivity norm and high cohesion.

Thank you!

Reviewer 4 Report

Dear authors,

the paper is very interesting. I suggest the next improvements:

1. The introduction section is too long. I suggest to include additional paragraph "2. Literature and hypothesis development" starting before subsection "Types and criteria".

2. The tables in the text must be improved according to the requirements of the journal.

3. The conclusion section must be expanded.

4. I suggest the subsections from the Discussion to be transferred in the Conclusion and rewrite the Discussion section giving strong argumentation for the obtained results based on a literature review.

Author Response

The authors express their deep gratitude to the referee for the thorough analysis of the manuscript, and hope that the answers to the comments (please, see below), as well as the corresponding adjustments to the manuscript itself are satisfactory.

  1. The introduction section is too long. I suggest to include additional paragraph "2. Literature and hypothesis development" starting before subsection "Types and criteria".

Response: Thank you for the suggestion. When preparing the Introduction section, we tried to adhere to the traditional structure (statement of the problem, the purpose of the study and a summary of the literature review sections) and keep the text to a necessary minimum. Partly in response to the other two reviewers’ recommendations, we made some additions to the Introduction section. We are concerned that shortening the text in the Introduction section could lead to the loss of important information. However, in the last paragraph, we removed a few sentences.

We added the section “Literature review and hypothesis development”.

  1. The tables in the text must be improved according to the requirements of the journal.

Response: We corrected decimal points separators (comas – to dots) in Table 1, when necessary.

  1. The conclusion section must be expanded.

Response: We expanded the “Conclusion” section by including into it the subsections “Theoretical and Practical Implications” and “Limitations and Future Research”.

  1. I suggest the subsections from the “Discussion” to be transferred in the “Conclusion” and rewrite the “Discussion” section giving strong argumentation for the obtained results based on a literature review.

Response: Thank you for the suggestion. As stated above, the subsections “Theoretical and Practical Implications” and “Limitations and Future Research” were moved from the “Discussion” section to the Conclusion section.

We also slightly expanded the “Discussion” section. Specifically, we tried to strengthen our argumentation by linking it more closely to the existing research literature, whenever such sources pertaining to the key aspects of our study could be located.

Thank you!

Reviewer 5 Report

I would like to thank the authors for the study carried out, I think that the subject is interesting and can provide relevant information to be able to understand even more, the psychological processes that influence the groups and subgroups of work.

Below, I have some observations in the different sections of this article, which I hope will be useful to improve your research.

Summary does not describe the sample, comments on the number of participating groups, but does not specify the area of the organization to which these work teams belong.

Introduction: 

*page 1, line 49, the quote from (Hill & Villamor, 2022); has different font sizes.

*The authors review cohesion studies but do not define it at the beginning. It is advisable to include the definitions of the variables of the study at the beginning, in order to understand the objective of the study more clearly. The authors do not define what they mean by productivity standards, performance effectiveness, and social effectiveness, which makes reading difficult, as there is no common thread between the variables of the study and what they want to do. 

*On page 2, second paragraph, the authors support the effectiveness of subgroups, but there is a lack of substantiation and/or examples to be able to understand the idea of the proposals. There is an important gap between this paragraph and the following one, where they try to propose the line of research and objectives, the authors say why the study is important, but they do not clarify what contributions their proposal will have, a proposal that personally has not been clear to me in the introduction.

* What is the theoretical framework on where this research is based? It is not developed in the introduction and it is not reflected in the conclusions either.

Method: 

*Sample: in which country was this study carried out? the 39 work teams to how many companies belong?

*Measurement: I have not seen citations of the measurements used for this study, are they self-constructed by the authors, if so, this should be clarified.

*The authors comment that the productivity norm variable of work groups and informal subgroups was assessed by means of a questionnaire, which contains five items formulated as proverbial sayings that reflect work-related norms such as the pace and quantity of work performed, although examples have been cited, I find it difficult to understand the measure they have used, why was it decided to use "proverbial sayings"? what is the scientific basis for this measure? I am sorry, but it is not clear to me.

*The authors collected gender as sociodemographic data, have they not considered including this data as a control variable? I think it would be interesting for the study, I do not expect to include this analysis, I just make a reflection out loud.

Author Response

The authors express their deep gratitude to the referee for the thorough analysis of the manuscript, and hope that the answers to the comments (please, see below), as well as the corresponding adjustments to the manuscript itself are satisfactory.

Abstract:

  1. Summary does not describe the sample, comments on the number of participating groups, but does not specify the area of the organization to which these work teams belong.

Response: In the Abstract, we briefly indicated the categories of participating organizations. We would readily describe the selection in more detail, but it would mean having to abbreviate some (perhaps, more important) details, since the Abstract should not exceed 200 words according to the Journal’s rules (already slightly exceeded by this manuscript).

Introduction: 

  1. page 1, line 49, the quote from (Hill & Villamor, 2022); has different font sizes.

Response: Thank you for pointing our attention to the issue. It has been corrected.

  1. The authors review cohesion studies but do not define it at the beginning. It is advisable to include the definitions of the variables of the study at the beginning, in order to understand the objective of the study more clearly. The authors do not define what they mean by productivity standards, performance effectiveness, and social effectiveness, which makes reading difficult, as there is no common thread between the variables of the study and what they want to do. 

Response: Thank you for the comment. We followed this particular logic to identify the key constructs as they are discussed in detail in the article. In the “Types and Criteria of Group Effectiveness” subsection, based on the literature review, we named the two types of effectiveness – performance effectiveness and social effectiveness – and showed that they are multidimensional constructs that include several identifiction criteria and performance indicators.

For example, we described the following four criteria for depicting performance effectiveness: (a) task effectiveness (reflects the degree of compliance of the results achieved by the group with the tasks set, for example, the amount of work performed, the number of completed and not completed tasks, the accuracy of task completion, the results for the current period relative to the results for previous periods); (b) process effectiveness (reflects the temporal and/or structural components of the work, for example, meeting the deadlines for completing tasks or projects, solving emerging problems or eliminating/mitigating emergency situations, maintaining the sequence of all operations in accordance with given standards); (c) quality effectiveness (reflects how both the results and work processes meet quality standards, for example, the quality of services provided or products produced); and (d) efficiency or cost-effectiveness (reflects the ratio of result achieved to the associated costs, or of planned and actual costs).

In our view, offering a single short definition of such a complex construct could be a potentially misleading oversimplification.

As for the group cohesion, it was defined in the beginning of the “Cohesion and Effectiveness of a Group/Subgroup” subsection. This subsection also offers a brief literature review on the topic of group cohesion and its connection to group effectiveness.

Similarly, we defined the “productivity norm” construct in the beginning of the “Productivity Norms and Group/Subgroup Effectiveness” subsection.

  1. On page 2, second paragraph, the authors support the effectiveness of subgroups, but there is a lack of substantiation and/or examples to be able to understand the idea of the proposals. There is an important gap between this paragraph and the following one, where they try to propose the line of research and objectives, the authors say why the study is important, but they do not clarify what contributions their proposal will have, a proposal that personally has not been clear to me in the introduction.

Response: Thank you for your comment. At the appropriate place in the article, we have inserted a small segment of text to clarify the idea that an informal subgroup is a collective actor that has an inherent effectiveness. In the subsection “Informal Subgroups in the Work Group”, we also expanded on this idea.

  1. What is the theoretical framework on where this research is based? It is not developed in the introduction and it is not reflected in the conclusions either.

Response: Next the Introduction section we added the “Theoretical Framework” section in order to present the conceptual grounding of the study – the micro-group theory with the corresponding references. Also within the “Conclusion” section (subsection “Theoretical and Practical Implications”) we briefly outlined potential contribution of the study to the further development of the micro-group theory.

Method

  1. Sample: in which country was this study carried out? the 39 work teams to how many companies belong?

Response: Thank you. In the corresponding segment of the Method section we specified the country and the number of participation organizations.

  1. Measurement: I have not seen citations of the measurements used for this study, are they self-constructed by the authors, if so, this should be clarified.

Response: Subsection Measures contains the reference to the GROUP PROFILE software package (Sidorenkov & Pavlenko, 2015). This source provides sufficient description of the computer program in general and of the specific questionnaires it features, including those used in the current study. So, we decided it should be sufficient and would not require references for each individual questionnaire.

  1. The authors comment that the productivity norm variable of work groups and informal subgroups was assessed by means of a questionnaire, which contains five items formulated as proverbial sayings that reflect work-related norms such as the pace and quantity of work performed, although examples have been cited, I find it difficult to understand the measure they have used, why was it decided to use "proverbial sayings"? what is the scientific basis for this measure? I am sorry, but it is not clear to me.

Response: Proverbial sayings – as concise expressions of folk wisdom – could be very helpful in providing quite accurate explanations of nearly any real life phenomenon. They are also very familiar to individuals with the shared cultural experiences across educational levels and professional specializations.

The group productivity norm questionnaire includes those proverbs/sayings that are very well known and well understood in the Russian-speaking environments, and are often used in everyday life and at work. As we understand, they were chosen by the developer of the questionnaire in order to mask the social desirability of the content of the items and, thereby, increase the reliability of the results.

  1. The authors collected gender as sociodemographic data, have they not considered including this data as a control variable? I think it would be interesting for the study, I do not expect to include this analysis, I just make a reflection out loud.

Response: Thank you for these interesting considerations. Our task was to focus specifically on informal subgroups, their own effectiveness and contribution to the effectiveness of the entire work groups – regardless their gender and/or other socio-demographic composition (the overall number of samples and their diversity supposedly balanced potential influence of these variables in this study). However, we do acknowledge that in the upcoming research, when the major issues with regard to informal subgroups are better understood, more refined research questions, involving these variables could be posed.  

Thank you!

Round 2

Reviewer 5 Report

Thank you for your response.